# Endometriosis: Molecular Pathophysiology and Recent Treatment Strategies—Comprehensive Literature Review

**DOI:** 10.3390/ph17070827

**Published:** 2024-06-24

**Authors:** Marcin Sadłocha, Jakub Toczek, Katarzyna Major, Jakub Staniczek, Rafał Stojko

**Affiliations:** 1Department of Gynecology, Obstetrics and Oncological Gynecology, The Medical University of Silesia in Katowice, Markiefki 87, 40-211 Katowice, Poland; streetbike5@wp.pl (J.T.); rstojko@sum.edu.pl (R.S.); 2Department of Neonatology, Municipal Hospital in Ruda Śląska, Wincentego Lipa 2, 41-703 Ruda Śląska, Poland; m.kasia88@gmail.com

**Keywords:** endometriosis, clinical diagnosis, pharmacological treatment, estrogen

## Abstract

Endometriosis is an enigmatic disease, with no specific cause or trigger yet discovered. Major factors that may contribute to endometriosis in the pelvic region include environmental, epigenetic, and inflammatory factors. Most experts believe that the primary mechanism behind the formation of endometrial lesions is associated with Sampson’s theory of “retrograde menstruation”. This theory suggests that endometrial cells flow backward into the peritoneal cavity, leading to the development of endometrial lesions. Since this specific mechanism is also observed in healthy women, additional factors may be associated with the formation of endometrial lesions. Current treatment options primarily consist of medical or surgical therapies. To date, none of the available medical therapies have proven effective in curing the disorder, and symptoms tend to recur once medications are discontinued. Therefore, there is a need to explore and develop novel biomedical targets aimed at the cellular and molecular mechanisms responsible for endometriosis growth. This article discusses a recent molecular pathophysiology associated with the formation and progression of endometriosis. Furthermore, the article summarizes the most current medications and surgical strategies currently under investigation for the treatment of endometriosis.

## 1. Introduction

Endometriosis is often regarded as a persistent condition marked by the abnormal growth of tissue resembling the lining of the uterus (endometrium) outside the uterine cavity [1,2]. It impacts an estimated 10–30% of women in their childbearing years, leading to severe chronic pelvic pain, infertility, and a decline in quality of life, affecting social and psychological well-being [3]. The underlying mechanisms of endometriosis are still unclear, but it involves various biological processes, including epigenetic changes and environmental factors. The abnormal endometrial tissue can survive and attach itself to the peritoneal membrane, aided by the activation of specific adhesion molecules and their receptors, overproduction of metalloproteinases, and activation of plasminogen [4]. This ectopic tissue exhibits unregulated growth due to altered steroid receptor function and stimulates new blood vessel formation [5,6]. Immune response irregularities and immune system dysfunction are key in the initial placement and ongoing survival of this abnormal tissue. Oxidative stress, defined by an imbalance in reactive oxygen species (ROS), is also a significant factor in the disease’s development and progression [7,8]. Endometriotic lesions are categorized into three types: superficial peritoneal surface lesions, ovarian lesions (also known as chocolate cysts or ectopic endometrial gland cysts), and deep-infiltrating lesions typically located in the rectovaginal area. While these are referred to as pelvic endometriotic lesions, there are also extrapelvic endometriotic lesions, which can be found in abdominal organs and, in rare cases, in the lungs [9]. Disease staging considers the location and severity, using systems like the American Society for Reproductive Medicine (ASRM) and the ENZIAN score, focusing on lesion location in the minor pelvis [10,11]. Treatment options include surgical removal of lesions and hormonal therapies to decrease endometrial growth, which are effective for pain relief and reducing the risk of recurrence but not recommended for treating infertility due to their impact on ovarian function. Surgical intervention may improve fertility, and assisted reproductive technologies like in vitro fertilization (IVF) are often the best approach for infertility linked to endometriosis [12,13]. Nevertheless, there is an increasing need for novel treatments that not only address symptoms and infertility but also lower recurrence and slow disease progression. Strategies targeting oxidative stress are emerging as potential treatments, given its significant roles in disease development and lesion progression.

Despite the prevalence of endometriosis, the exact molecular and cellular mechanisms underlying endometriosis remain poorly understood. Current treatment options, which include medical and surgical therapies, often fail to provide a permanent cure, with symptoms frequently recurring after the discontinuation of medication. A detailed exploration of the molecular pathophysiology of endometriosis is essential to uncover the potential biomedical targets that could lead to more effective and long-lasting treatments. This publication addresses the critical gaps in the understanding of endometriosis at a molecular level and evaluates the latest treatment strategies.

## 2. Aim

The aim of this publication is to provide a comprehensive review of the molecular pathophysiology of endometriosis and to evaluate recent advancements in treatment strategies, with a focus on understanding the underlying mechanisms and identifying potential new therapeutic targets.

## 3. Methods

The search strategy focused on identifying studies that expound the underlying molecular mechanisms, explore novel therapeutic approaches, and assess the efficacy of existing treatment modalities. Our literature review on endometriosis ensured a thorough examination of existing research and the identification of key studies that address the various biological, clinical, and therapeutic dimensions of the condition. In the context of endometriosis research, a literature search ensures the inclusion of relevant studies, minimizes selection bias, and provides a foundation for synthesizing evidence on the pathophysiology, diagnosis, and management of the disease. This section outlines the search strategy employed to gather literature on the molecular pathophysiology and treatment strategies for endometriosis, adhering to the principles of the Scale for the Assessment of Narrative Review Articles (SANRA).

### 3.1. Article Selection Process and Selection Criteria

The search strategy involved multiple electronic databases to capture scientific literature. The following databases were utilized: PubMed, Web of Science, Scopus, Cochrane Library, Medline, and Google Scholar. Keywords and Medical Subject Headings (MeSH) terms used in the search included: endometriosis, molecular pathophysiology, etiological factors, environmental factors, epigenetic modifications, inflammatory response, retrograde menstruation, endometrial lesions, angiogenesis inhibitors, immunomodulators, epigenetic modifiers, treatment strategies, medical therapies, and surgical interventions. Boolean operators (AND, OR, NOT) were employed to refine the search results and ensure the inclusion of all relevant studies.

The search was filtered to include articles published between 2010 and 2024. The titles and abstracts of the identified articles were reviewed to determine their relevance to the topic. The full texts of 194 potentially relevant articles were examined to ensure they met the inclusion criteria. A total of 107 articles were included in the review.

### 3.2. Inclusion Criteria

Studies published in peer-reviewed journals;Articles in English;Research on the molecular mechanisms of endometriosis;Studies discussing current and emerging treatment strategies;Reviews, meta-analyses, clinical trials, and observational studies.

### 3.3. Exclusion Criteria

Non-peer-reviewed articles;Studies not focused on endometriosis;Articles in languages other than English;Publications older than 20 years, unless they were seminal works.

### 3.4. Synthesis of Findings

The extracted data were synthesized to provide a comprehensive overview of the molecular pathophysiology of endometriosis and current and emerging treatment strategies.

## 4. Pathophysiology of Endometriosis

### 4.1. Inhibiting RAF/MEK/ERK Signal Pathway by Targeting RKIP Reduce Progression of Ectopic Lesions

It is important to discover the molecular pathways involved in the activation of intracellular enzymes, leading to increased phosphorylation. Furthermore, it is important to discover the phenotype of these specific cells, known as hEM15A cells, which have some important characteristics that resemble cancer cells [14,15,16,17]. This is why current research is focused on non-invasively targeting and diagnosing the RAF/MEK/ERK signaling pathway for future treatment options. Signaling pathways, primarily MAPK/ERK, PI3K/Akt, and NF-κB, are responsible for the regulation of apoptosis in endometriotic cells [18,19,20]. Activation of ERK1/2, which acts as an anti-apoptotic protein in both eutopic and ectopic endometrial glands throughout the menstrual cycle, leads to various cellular metabolic changes. Non-estradiol (E2)-targeted therapies, such as selective PGE2 inhibitors, inhibit PGE2 (EP) 2 and EP4 receptors, facilitating apoptosis through the activation of multiple pathways, including ERK1/2, AKT, and NF-κB [21,22,23]. These specific pathways accelerate the apoptosis process by approximately 50% in both epithelial and stromal cells. Therefore, due to their effectiveness as selective or combination inhibitors, these therapies are ideally suited for the treatment of stage I and II endometriosis. PGE2 inhibitors are preferred due to their fewer side effects and lack of hypoestrogenic effects. In endometriosis, MAPK is activated, initiating intracellular transmission of extracellular signals and entailing cellular processes associated with high levels of phosphorylated extracellular signal-regulated kinases [1,24,25,26]. RAS attached to RAF affects the activation of ERK, leading to phosphorylation of ERK, which is a major component of the MAPK signaling pathway. cERK1/2 regulates the expression of c-fos and c-jun to regulate mitotic processes and endometrial cell viability. Estradiol (E2), IL-1β, and TNF-α stimulate ERK1/2 phosphorylation in endometriotic stromal cells but not in normal endometrial cells [27,28,29,30,31,32]. Three families of MAPK signaling kinases are identified as extracellular signal-regulated kinase (ERK), p38, and c-Jun N-terminal kinase (JNK). All three pathways are stimulated by components of the extracellular environment, with ERK predominantly being activated by inflammation and growth factors, while JNK and p38 are mainly activated by stress and inflammation [33,34,35]. It is posited that the MMP/TIMP signaling pathway plays a critical role in the formation of metastases and the aggressive invasion of cancer. MMPs, a group of proteolytic enzymes, break down the extracellular matrix, thus facilitating the proliferation of cellular metastases [1]. Conversely, TIMPs serve to block the breakdown activity of MMPs. In response to proinflammatory stimuli, stromal cells primarily increase Interleukin (IL)-33 [36]. This increase in IL-33 augments the expression of MMP-9 in hOVEN-SCs through the ST2/MAPK pathway, enhancing their capability to invade [37]. Additionally, TNF-α is known to activate MMP9, leading to the conversion of SRC-1 into its 70 kDa isoform. This C-terminal variant of SRC-1 acts to shield immortalized human endometrial epithelial cells (IHEECs) from TNF-α-induced apoptosis by inhibiting the activation of procaspase 8, thereby preventing its use [38,39]. One of the most promising pharmaceuticals is currently Sorafenib, which has undergone phase IV clinical trials in several types of cancer and significantly inhibits RAF kinase phosphorylation by 64% via the MAPK/ERK cascade in the stromal cells of endometriosis patients [40]. Another drug, vemurafenib, is approved by the FDA for the treatment of metastatic melanoma and significantly inhibits ERK phosphorylation by more than 60% in endometriotic stromal cells and epithelial cells simultaneously. Some of the MEK1/2 inhibitors can increase the expression level of progesterone receptor (PR)-αβ in endometriotic stromal cells [41,42,43]. However, it should be remembered that despite promising clinical results, the above-mentioned methods of treating endometriosis with MAPK inhibitors are not free from side effects such as limitation of reproductive functions, including inhibition of ovulation, embryotoxicity, and teratogenicity [44].

### 4.2. Increased Levels of miRNAs Favor Survival Feature and Lead to Progesterone Resistance

Elevated levels of certain miRNAs, specifically three, have been identified to influence the expression and function of progesterone receptors, contributing to progesterone resistance, a critical factor in the pathophysiology of endometriosis [45,46,47,48]. The miRNAs implicated in this mechanism include miR-29c-3p, miR-126-3p, and miR-143-3p, with a particular focus on miR-143-3p due to its notable presence in the circulation and ectopic lesions of endometriosis patients [49,50,51]. Notably, miR-29c has been consistently observed to increase in lesion tissues compared to normal endometrial tissue across three different studies [52,53,54,55]. Among these, a study by Hawkins and colleagues [56] stands out, as it uniquely explored the functional implications of overexpressed and differentially expressed miRNA [57]. In the context of miR-29c, which is significantly overexpressed depending on the type of tissue, research involving primary human endometrial stromal cells cultured in vitro has been conducted. It was found that extracellular matrix proteins, such as COL7A1, UPK1B, and TFAP2C, which are likely targets of miR-29c, showed reduced expression in cells where miR-29c mimic was overexpressed. Additionally, various studies have confirmed a direct impact on the 3′-UTR of these genes. Consequently, the unusual increase of miR-29c in ovarian lesions, including chocolate cysts, is functionally linked to the altered expression of extracellular matrix proteins, which are a characteristic of endometriosis, particularly in isolated endometrial stromal cell cultures [58]. Recent work by Long [59] examined the expression of the miR-29 family (miR-46, miR-29a, miR-29b, and miR-29c) in the endometrial layers of women without endometriosis and compared it with samples from both ectopic and eutopic endometrial cells. Contrary to earlier studies [60], they noted a reduction in miR-29c expression in tissue lesions, although they did not specify whether the lesion type was ovarian chocolate cysts or peritoneal lesions. Further in vitro studies using the CRL-7566 endometriosis cell line demonstrated that increased miR-29c expression correlated with decreased cell proliferation and invasion, as well as increased apoptosis. In this mechanism, the regulatory effect is attributed to c-Jun kinases, a group of protein kinases integral to stress signaling pathways that influence gene expression, neuronal plasticity, regeneration, cell death, and cellular aging. It is important to recognize that while these findings indicate a reduction in miR-29c expression at the sites of endometriotic lesions, potentially leading to increased cell proliferation and invasion due to the low miR-29c levels, other studies have consistently shown an increase in miR-29c expression [61].

### 4.3. Dysregulation of Immune System Concerning the Roles of Macrophages and Treg Cells

(a) Macrophages are the key cells responsible for enhancing inflammation and following with neutrophil diapedesis through the release of certain products called chemokines [62]. These two types of cells are especially observed in abundant numbers within endometrial lesions. Furthermore, the phagocyte activity of peripheral monocytes and neutrophil granulocytes seems to be decreased, and this activity has been influenced by the presence or removal of endometriotic lesions in women with endometriosis [63]. Macrophages play a crucial role in the formation of endometrial lesions and concurrent inflammation. Nevertheless, the general number of monocytes in the peripheral blood of endometriosis patients was not altered [13]. In the peritoneal fluid, they are the most plentiful cells found in healthy individuals; therefore, it is not surprising that their increased concentration has been observed in local endometriosis environments. During the development of endometrial implants, the number of these cells significantly increases [64]. A meaningful increase in the number of macrophages has also been shown in the eutopic endometrium in women with endometriosis. Further investigations have shown that macrophages in endometriosis patients are not fully functional [65]. The decrease in the expression of CD3 (cluster of differentiation 3) and annexin A2 is linked with the deterioration of phagocytic function. Nevertheless, the production of inflammatory mediators by macrophages promotes the implantation process and the proliferation of endometrial cells, resulting in the development of endometrial lesions [66]. Macrophages were also identified to concurrently occur in increased numbers with nerve fibers in peritoneal endometrial lesions. It has been hypothesized that macrophages and nerve fibers interact to promote the pain symptoms associated with endometriosis [67]. It is worth noting that Greaves et al. recently revealed that estradiol is a potent mediator in the interactions between macrophages and nerve tissue in peritoneal endometriosis [68].

(b) Treg cells: The etiopathogenesis of autoimmune or autoinflammatory disorders may be related to the incoherent function of CD4^+^CD25^high^FOXP3^+^ regulatory T cells (Treg cells) [69]. One of the main functions of Treg cells is to inhibit the effector responses of T cells, macrophages, and natural killer cells. Furthermore, they interpose anti-inflammatory responses and lead to the production of various proinflammatory cytokines [70]. Hence, Treg cells are considered pivotal mediators to gather immune tolerance. Moreover, they seem to be responsible for the suppression of anti-tumor immunity and the concomitant promotion of tumor growth. The phenotype and main function that Treg cells rely on is the high constitutive expression of CD25, which is a subunit of the IL-2 receptor, which is responsible for their development and progression [71]. FOXP3 transcription factor determines their development and suppressive activity. Currently, it has been revealed that Treg cells may play a role in the immunopathogenesis of endometriosis. Elevated numbers of Treg cells and increased FOXP3 expression have been reported in the eutopic and ectopic endometria of women with endometriosis. Moreover, it has been revealed that the proportion of Treg cells may be meaningfully augmented in the milieu of peritoneal fluid of patients with endometriosis [72]. Interestingly, an increased level of Treg cells in the peritoneal fluid may be related with a decreased proportion of Treg cells in the circulating peripheral blood [73]. Enhanced numbers of Treg cells may influence the local peritoneal immune responses associated with NK cell suppression; hence, they can promote the survival of endometriotic cells and lead to the growth and invasion of endometriotic lesions [74]. Alterations in the proportions of Treg cells may also be related to increased autoimmune phenomena, which likely reveal an autoimmune phenotype of endometriosis. Thus, Treg cells appear to be significant contributors to the immunopathogenesis of endometriosis and may deserve special attention [75].

## 5. Surgical Treatment

Invasive treatments considering surgical intervention for eliminating endometriotic lesions and reducing or releasing adhesions have a crucial impact on the management of endometriosis. Historically, surgical approaches were achieved with open surgery, but in recent decades, laparoscopy has dominated. Elimination or excision of endometriotic lesions may be induced by excision, a diathermy device, or ablation/vaporization. The main aim of reducing and releasing adhesions is to re-establish proper pelvic anatomy. Furthermore, some clinicians apply interruption of the pelvic nerve pathway with the intention of mitigating the intensity of the pain. The current section focuses on the efficacy and safety of surgical intervention for the management of pain in women with endometriosis. Technical guidance on surgical techniques for surgery in endometriosis has been previously published by a working group of ESGE, ESHRE, and WES [76].

### 5.1. Ablation vs. Excision

Most of the clinical trial investigations likewise reveal no relevant difference between the utilization of ablation and excision of endometriotic lesions, especially for minimal and mild endometriosis. One review examined the mean reduction in visual analogue scale (VAS) scores from baseline to 12 months postoperative, or mean VAS scores at 12 months postoperative, for dysmenorrhea, dyschezia, and dyspareunia and concluded that there were no significant differences between the excision and ablation groups with regards to improving the pain measured with the above parameters [77].

### 5.2. Peritoneal Endometriosis

Some consider superficial peritoneal endometriosis (SPE) as a completely different entity than ovarian lesions and deep endometriosis. Nevertheless, certain researchers argue that they are frequently diagnosed together and are likely to be different units of the same condition [78]. There are no targeted trials specifically focusing on the investigation of the effect of surgery for SPE on pain symptoms. In some particular studies, only women with stage I and II ASRM were included, and the majority of these may have had SPE [79]. On the other hand, stage I and II ASRM disease may also include women with ovarian endometriomas equal to or smaller than 1 cm or deep endometriosis; hence, it would be impossible to create as well as generalize the specific results of these studies for women with isolated SPE [80].

### 5.3. Ovarian Endometriosis

An isolated endometrial cyst is a very sporadic lesion as an isolated entity. In most cases, it is accompanied by foci of superficial peritoneal endometriosis or deep endometriosis. Therefore, when diagnosing or treating an endometrial cyst, it is recommended to thoroughly assess the pelvis and simultaneously treat all identified lesions. The recommended technique for removing an endometrial cyst is laparoscopic excision [81]. Compared to other techniques, which include drainage and coagulation, the excision technique is usually associated with a lower rate of cyst recurrence, repeated surgical intervention, and a less frequent comeback of dysmenorrhea and painful intercourse [82]. The resection of the endometrial cyst or a small fragment of it should always be examined from a histopathological angle to confirm the diagnosis and exclude a malignant nature of the lesion [83]. The technique of laser ablation of the cyst wall, or its electrocoagulation, makes it impossible to obtain material for histopathological examination. Laser vaporization is characterized by a higher rate of cyst recurrence within twelve months compared to cystectomy, but within a five-year period, the frequency is comparable [84]. Regardless of the method used for surgical treatment of endometrial cysts, the main priority should always be to minimize the impact on core ovarian tissue and, thus, on the ovarian reserve [79]. In the event of endometrial cyst surgery, the patient should receive pivotal information concerning the reduction in ovarian reserve—fertility deprivation—especially if the cyst consists of a pretty large diameter, likewise occurs bilaterally, or is recurrent. Surgical treatment of endometrial ovarian cysts in patients planning for pregnancy should be carried out with the greatest possible care in preserving the ovarian cortex tissue and at the same time removing blocked fallopian tubes. Operative consent to surgery should always include information summarizing the risk of losing the ovary or ovaries and the consequences of losing them. For patients who are considering becoming pregnant in the future, pre-operative assessment of the ovarian reserve [AMH—anti-Müllerian hormone test] should be considered and the possibility of pre-operative egg collection for storage should be discussed, especially if endometrial cysts occur on both sides [85]. Among benign ovarian tumors, the presence of endometrial cysts has the most significant impact on reduction in the ovarian reserve [86].

### 5.4. Deep Endometriosis (DE)

Excision of deep endometriosis (DE) nodules aims to alleviate lesion-associated pain, though its effect on fertility remains a matter of debate. Surgeries, particularly complex ones, should be undertaken by specialists. ESHRE reports associate surgeries for deep lesions with significant rates of intraoperative and postoperative complications. CNGOF highlights potential risks, including anastomotic leaks, fistulas, and rectal dysfunction. In some instances, bladder atony might occur due to surgical disruption of the hypogastric plexus. ESHRE contrasts the outcomes of shaving versus segmental resection. Both ESHRE and CNGOF endorse procedures involving bladder excision. While all the aforementioned data reference it, only NICE and the German group advocate for pre-operative imaging using ultrasound or MRI. In February 2020, ESGE/ESHRE/WES issued guidelines addressing the surgical management of deep-infiltrating lesions [87]. DE is distinguished by its severe clinical manifestation, which is characterized by ectopic endometrial tissue penetrating more than 5 mm beneath the peritoneal surface, leading to localized inflammation, fibrosis, and muscle-layer hyperplasia [88]. For describing the DE lesions during laparoscopic procedures, it is necessary to adjust them to the ENZIAN score. This particular score describes endometriotic lesions according to diameter, localization, and organ lesions [20]. Typically presenting as a multifocal condition, DE affects various pelvic locations, including the pouch of Douglas, uterosacral ligaments, pelvic nerves, rectum, urinary bladder, and ureters [89]. The specific pathophysiological changes are largely unknown and may vary over time without a predictable progression of endometriotic lesions. Women with endometriosis may suffer from symptoms like painful menstruation, dyspareunia, painful urination, constipation, chronic pelvic pain, and infertility. However, the specific pain types and implant locations vary widely, and some with minimal disease may experience severe pain and infertility, while others with extensive lesions might have minimal symptoms [90]. The disease’s heterogeneity and uncertain pathogenesis complicate its diagnosis. Historically, laparoscopic visualization followed by histological confirmation of lesions was the diagnostic standard. Yet laparoscopy is not without risks and may not always clearly reveal deep endometriosis, particularly when involving retroperitoneal structures like the ureters and nerve roots, leading to potential misdiagnoses and delayed treatment [91]. The International In-depth Endometriosis Analysis (IDEA) consensus promotes a systematic ultrasound approach to enhance the detection of pelvic lesions [92]. MRI is also recognized for its accuracy in evaluating DE, especially when the rectum, ureters, and nerve roots are involved, and is increasingly used to monitor anatomical responses to treatment and distinguish between endometriosis and adenomyosis, the latter being a distinct and varied condition characterized by the invasion of endometrial tissue into the myometrium. Adenomyosis, which may occur independently or in conjunction with DE, is found in approximately 30% of cases [93].

### 5.5. Interruption of Pelvic Nerve Pathways: LUNA and PSN

The effectiveness of laparoscopic interruption of certain pelvic nerve pathways considering primary and secondary dysmenorrhea was analyzed in a Cochrane review on women with endometriosis [94]. RCT investigations estimate the effect of laparoscopic uterosacral nerve ablation (LUNA) as well as gathering an account of the conservative laparoscopic surgery for endometriosis, evaluating the effects of presacral neurectomy (PSN) (in a laparoscopic way, likewise laparotomic) [95]. The RCT studies on the LUNA procedure reveal that this technique did not offer any supplementary benefit together with conservative intervention one year after surgery. An evaluation at 6 months after the primary surgery did not show any additional benefit either. There were significant benefits of PSN at 6 months (1 RCT) and 12 months (2 RCTs) [96]. One of the RCTs comprised the Cochrane investigation, which included the 24-month follow-up outcomes of PSN in addition to laparoscopic surgery for endometriosis compared to the laparoscopic procedure associated only with the treatment of severe dysmenorrhea, dyspareunia, and pelvic pain due to endometriosis [97]. The frequency and severity of dysmenorrhea, dyspareunia, and CPP and quality of life were evaluated. The PSN group had much better mitigation of dysmenorrhea, dyspareunia, pelvic pain, and quality of life compared to laparoscopic surgery only. Furthermore, frankly speaking, PSN has been associated with an increased risk of adverse effects such as bleeding, constipation, urinary urgency, and a painless first stage of labor. Endometriosis surgery has improved in recent decades, and the place of PSN needs to be confirmed for patients who undergo radical excision of deep endometriosis [98].

## 6. Medical Approach

### 6.1. GnRH Antagonist and Agonist

Medicaments which belong to the group of gonadotropin analogues (previously, solely GnRH agonists were available) are much more effective than placebo groups in alleviating pain symptoms in endometriosis due to the rapid atrophy of the endometrium, incorporating the ectopic endometrial foci [99]. There is no certain alternative when it comes to the route of administration of GnRH agonists or a route that does not affect side effects. GnRH agonist therapy has been associated with abundant side effects, such as vaginal dryness, hot flushes, weight gain, acne, and headaches [100]. The risk related to decreased bone density becomes much higher when the dose of the drug becomes higher. In this case, there is greater safety of doses of 1.88 mg over 3.75 mg, with an additional issue linked to pain alleviation [101]. In order to mitigate the effect of GnRH agonists, especially on bone density structure, you can add add-back therapies, which comprise the following:Progestogen—norethisterone octate;Combination of estrogen and progesterone;Selective estrogen receptor modulator (SERM);Bisphosphonates or testosterone [102].

Add-back therapy mitigates to the reduction in bone mass density, particularly in the lumbar spine, during the use of GnRH agonists without reducing their effectiveness in alleviating the pain outcomes of endometriosis [103]. With comments on the action of GnRH agonists, they are second-line drugs in the case of endometriosis in the event of failure of combined therapy with contraceptive pills. Caution should be introduced in their use in adolescent patients with concerns about bone loss, who may be at risk during puberty [104].

Clinical trials have also included the effectiveness of GnRH antagonists such as elagolix, relugolix, and linzagolix in the treatment of pain for patients with endometriosis. Elagolix has been used in the USA for the treatment of endometriosis in two therapeutic doses of 150 and 200 mg [105]. Higher doses of GnRH antagonists are better at fighting endometriosis but are used with frequent side effects with a profile comparable to GnRH agonists. Relugolix administered orally at 10, 20, and 40 mg relieves endometriosis pain in a dose-independent manner. Oral relugolix (40 mg GnRH antagonist + 1 mg estradiol + 0.5 mg norethindrone acetate) made available for 24 weeks was more effective than a placebo in relieving endometriosis pain at the initiation of therapeutic measures [106].

### 6.2. Selective Progesterone Receptor Modulators (SPRMs)

SPRMs, or selective progesterone receptor modulators, are a class of progesterone receptor ligands that exhibit varied actions as agonists, antagonists, or mixed agonists/antagonists based on the specific tissue they target. These medications effectively halt ovulation without inducing the systemic effects associated with estrogen deprivation, as they do not affect estradiol secretion and maintain circulating estradiol levels within the normal range [2]. SPRMs also play a role in curbing endometrial proliferation, reducing endometrial bleeding by directly influencing endometrial blood vessels, and lowering prostaglandin production in the endometrium [107]. The human endometrium, expressing both PR isoforms throughout the menstrual cycle, undergoes changes in receptor quantity and the PRA-to-PRB ratio in response to fluctuating ovarian steroid levels, generally showing a higher presence of PRA [108]. These receptors are predominantly found in the glandular epithelium compared to the endometrial stroma, with the highest expression of PRA and PRB occurring mid-cycle and gradually decreasing to the lowest levels during the late secretory phase. In the secretory phase, stromal PR specifically localizes in decidualization cells, a key process in endometrial differentiation that is crucial for embryo implantation [109]. Activation of stromal PR also triggers paracrine actions that inhibit epithelial proliferation. Although splice variants of the PGR transcript and a potential truncated isoform (PRC) have been identified in the human endometrium, their exact roles are yet to be fully understood [110].

For treating endometriosis, drugs such as Ulipristal acetate (UPA), telapristone acetate, vilaprisan, and tanaproget are recommended. Generally, SPRMs are well received, with typical side effects including headaches, abdominal pain, nausea, dizziness, and menometrorrhagia, as well as the fact that chronic use can cause liver toxicity. Mifepristone and asoprisnil are among the most thoroughly studied SPRMs in this category [111]. The impact of UPA on endometriosis lesions and symptoms was observed over a 27-month period leading up to surgery, with 58% of cases showing progesterone receptor modulator-associated endometrial changes (PAECs) in both eutopic endometrium and ectopic lesions, correlating with decreased pain and amenorrhea. However, more data are needed to conclusively determine their safety and efficacy [112]. Figure 1 represents the different experimental treatments for endometriosis with their target sites. Kisspeptin/neurokinin B/dynorphin (KNDy) neurons with their hypothalamic connections were also identified as a potential target [113,114].

## 7. Conclusions

Endometriosis is a complex and enigmatic disease that continues to pose significant challenges in terms of understanding its etiology, diagnosis, and treatment. This paper has explored various aspects of endometriosis, including its pathophysiology, surgical treatment options, and emerging medical therapies. Despite the ongoing research efforts, the exact cause of endometriosis remains elusive, and the disease’s impact on the lives of affected individuals is substantial.

The pathophysiological mechanisms discussed in this paper shed light on the intricate processes involved in the development and progression of endometriosis. These processes include the dysregulation of signaling pathways, such as the RAF/MEK/ERK pathway, and the role of micro-RNAs in promoting survival and progesterone resistance. Understanding these mechanisms is critical for the development of targeted therapies aimed at addressing the root causes of the disease. Surgical treatment options for endometriosis have been a mainstay in managing the disease, with laparoscopic procedures being favored for their minimally invasive nature. However, the guidelines discussed in this paper highlight the importance of tailoring surgical interventions to the specific needs of each patient, considering factors like lesion type and severity. Adhesiolysis and the treatment of deep-infiltrating endometriosis remain areas of debate, underscoring the need for individualized approaches. In recent years, novel medical approaches have emerged as promising alternatives to traditional treatments. GnRH antagonists, selective estrogen receptor modulators (SERMs), and selective progesterone receptor modulators (SPRMs) have shown potential in alleviating endometriosis-related symptoms while minimizing the systemic side effects associated with hormonal therapies. These emerging therapies offer new hope for individuals with endometriosis, particularly those who do not respond well to existing treatments.

## Figures and Tables

**Figure 1 pharmaceuticals-17-00827-f001:**
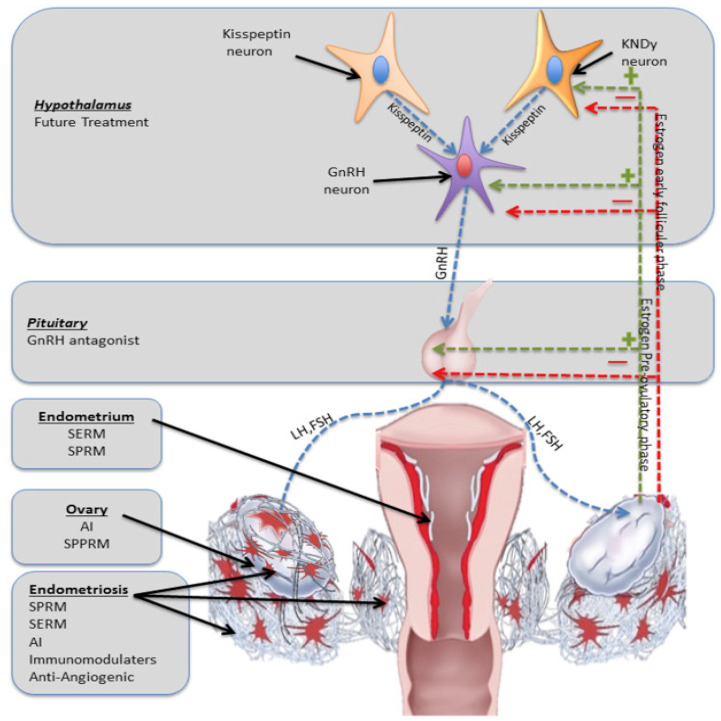
Schematic representation of the different experimental treatments for endometriosis with their target sites. Kisspeptin/neurokinin B/dynorphin (KNDy) neurons with their hypothalamic connections were also identified as a potential target by Bedaiwy for the future of endometriosis medical therapy and Fertil Steril 2017 [114].

## Data Availability

Not applicable.

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
