# Peer review of "Endometriosis: Molecular Pathophysiology and Recent Treatment Strategies—Comprehensive Literature Review"

_pharmaceuticals, 2024, doi:10.3390/ph17070827_

Round 1

Reviewer 1 Report

Comments and Suggestions for Authors

In the systematic literature review by Sadlocha et al., the authors gave an overview on the molecular pathophysiology and recent treatment strategies. What bothers me most is that many sections and assertions are given without references, for example lines 129-137, lines 245-265 and lines 338-355 are without any citation. Similarly in the lines 95-115 I would have welcomed to get more citations (e.g. lines 95, 99, 103, 108-110). 

Line 47 - You mean Pelvic endometriotic lesions, because there are also extra-pelvic endometriotic lesions.

Line 48 - Ovarian lesions are not only chocolate cysts, there are more cyst types but also ectopic endometrial glands in the ovary.

Another main thing which is disturbing is that only guidelines for endometriosis treatments are cited. Why? 

Comments on the Quality of English Language

Many abbreviations are not explained in the text or beneath the Figure such as HPO, which is also not in the list. 

Line 11 The itself can be omitted

Line 18 it is not other but instead additional factors

Line 39 Is it really an overproduction of MMPs or is it an activation by P4 withdrawal? Please cite the manuscripts by Marbaix and other colleagues.

Line 80 The sentence PGE2 inhibitors, especially COX-2, are preferrred is wrong or misleading.

Line 105 You cited ref 06, maybe this is a typo.

Lines 119/122 Why not shift the three miRs to line 119?

Author Response

Dear Reviewer,

Alteration List:

  1. In the abstract: I added an extra “additional factor” it is highlighted on yellow color. (It was recommended by the reviewer 1)
  2. In the introduction, I added some extra things concerning extra pelvic endometriotic lesions. It is also delineated by the yellow color 
  3. In the chapter pathophysiology, associated with endometriosis (I added an extra chapter related to dysregulation of the immune system)
  4. The figures are deleted ( both of them)
  5. In the chapter on surgical treatment, I’ve altered a few things concerning ablation vs excision, LUNA, and PSN. 
  6. Completely changed the medical approach for GnRH.
  7. Totally, alter the way of references in the article. ( highlighted by the yellow color)
  8. A few extra abbreviations are added as well as put to the list of abbreviations.

Sincerely Jakub Toczek

Reviewer 2 Report

Comments and Suggestions for Authors

Dear Authors,

The review entitled „Endometriosis: Molecular pathophysiology and treatment strategies – comprehensive literature review”  aims to overlook the pathophysiology of the disease from the molecular basics, as well as the treatment strategies including surgical and pharmaceutical approaches. The language of the article is clear in general.

My important general comment about the article is, that the authors have a habit to cite literatures in triplets or quadriplets, which is (in general) absolutely unnecessary in most of the presented cases, and for me unacceptably unscientific in such a number and as a tendency (3-5, 10-13, 14-17, 18-20, 21-23, 24-26, 27-29, 30-32, 33-35, 36-38, 39-41, 47-50, 51-53, 54-47, 60-62). Moreover, the majority of these triplets or quadruplets refer to one fact, while on the other hand there are major and large segments of the review, with a lot of cited facts without any references (line 245-268, with only one reference at the end, referring only to the MRI examination while there are a lot of other facts, which are mentioned in the paragraph- half a page). Other examples without the need for completeness lacking the reference: line 283 (ACOG guidelines, ESHRE guidelines are referred to but not cited), 302-305 pathomechanism of GnRH antagonists, 349, 352. Ref. 98 is a general review, citing the original article from the subject would be more relevant, this happens at other places as well.  

I specifically miss a few major and important facts from the review: Referring to the most recent ESHRE guideline for endometriosis from 2022, mentioning the liver toxicity of ulipristal, mentioning the importance of estrogen supply with GnRH anragonists, etc. The descriptions of surgical treatments are unclear and should be overlooked, etc. 52-55 surgical removal of lesions is not recommended due to their negative impact on ovarian reserve (true in some cases which underlines the importance of clear references, not in triplets…) while 180-182 “International guidelines now agree on recommending surgical approaches for women with presumed mild endometriosis and infertility issues.”- again without any reference.

In general, the article contains valuable information but in the current form could not be accepted. I suggest maor revision of the article, including the method of references. The surgical tretment should me much more precise and should include references from up-to-date guidelines. There are much more theories to the pathogenesis and pathophysiology of endometriosis than those the review targets, and these should also be included, e.g. genetic, epigenetic theory, the pathogenesis of deep endometriosis, stem cell theory, etc.).

Comments on the Quality of English Language

Only minor edition of English is required.

Author Response

(The authors gave the same response as above.)

Reviewer 3 Report

Comments and Suggestions for Authors

The paper is surely interesting. It reflects the scope of this Journal. Here my concerns:

In the abstract, I would add some results after the aim in order to give to the readers a direct data about this review

Do you think medical treatment can impact of fertility preservation or strategy to get pregnant? Discuss it (DOI: 10.1016/j.ejogrb.2020.09.045)

Nowadays, the literature deals with the role of dysregulation of immune system (macrophages and Treg cells) in the onset of endometriosis? What do you think?

Regarding DIE, you should mention ENZIAN Score (doi: 10.1007/s00404-022-06451-1)

What do you think about the tolerability of gonadotropin-releasing hormone analogues for the treatment of endometriosis? Discuss it (DOI:10.1080/17425255.2020.1789591)

Comments on the Quality of English Language

Minor revision by english mother tongue is needed.

Author Response

(The authors gave the same response as above.)
